# GraphMaker: Can Diffusion Models Generate Large Attributed Graphs?

**Mufei Li**                                    *mufei.li@gatech.edu*
*School of Electrical and Computer Engineering*
*Georgia Institute of Technology*

**Eleonora Kreačić**                            *eleonora.kreacic@jpmorgan.com*
*J.P. Morgan AI Research*

**Vamsi K. Potluru**                            *vamsi.k.potluru@jpmchase.com*
*J.P. Morgan AI Research*

**Pan Li**                                      *panli@gatech.edu*
*School of Electrical and Computer Engineering*
*Georgia Institute of Technology*

**Reviewed on OpenReview:** *https://openreview.net/forum?id=0q4zjGMKoA*

## Abstract

Large-scale graphs with node attributes are increasingly common in various real-world applications. Creating synthetic, attribute-rich graphs that mirror real-world examples is crucial, especially for sharing graph data for analysis and developing learning models when original data is restricted to be shared. Traditional graph generation methods are limited in their capacity to handle these complex structures. Recent advances in diffusion models have shown potential in generating graph structures without attributes and smaller molecular graphs. However, these models face challenges in generating large attributed graphs due to the complex attribute-structure correlations and the large size of these graphs. This paper introduces a novel diffusion model, GraphMaker, specifically designed for generating large attributed graphs. We explore various combinations of node attribute and graph structure generation processes, finding that an asynchronous approach more effectively captures the intricate attribute-structure correlations. We also address scalability issues through edge mini-batching generation. To demonstrate the practicality of our approach in graph data dissemination, we introduce a new evaluation pipeline. The evaluation demonstrates that synthetic graphs generated by GraphMaker can be used to develop competitive graph machine learning models for the tasks defined over the original graphs without actually accessing these graphs, while many leading graph generation methods fall short in this evaluation.

## 1 Introduction

Large-scale graphs with node attributes are widespread in various real-world contexts. For example, in social networks, nodes typically represent individuals, each characterized by demographic attributes (Golder et al., 2007). In financial networks, nodes may denote agents, with each node encompassing diverse account details (Allen & Babus, 2009). The task of learning to generate synthetic graphs that emulate real-world graphs is crucial in graph machine learning (ML), with wide-ranging applications. A traditional use case is aiding network scientists in understanding the evolution of complex networks (Watts & Strogatz, 1998; Barabási & Albert, 1999; Chung & Lu, 2002; Barrat et al., 2008; Leskovec et al., 2010). Recently, a vital application has emerged due to data security needs: using synthetic graphs to replace sensitive original graphs, thus preserving data utility while protecting privacy (Jorgensen et al., 2016; Eliáš et al., 2020). This

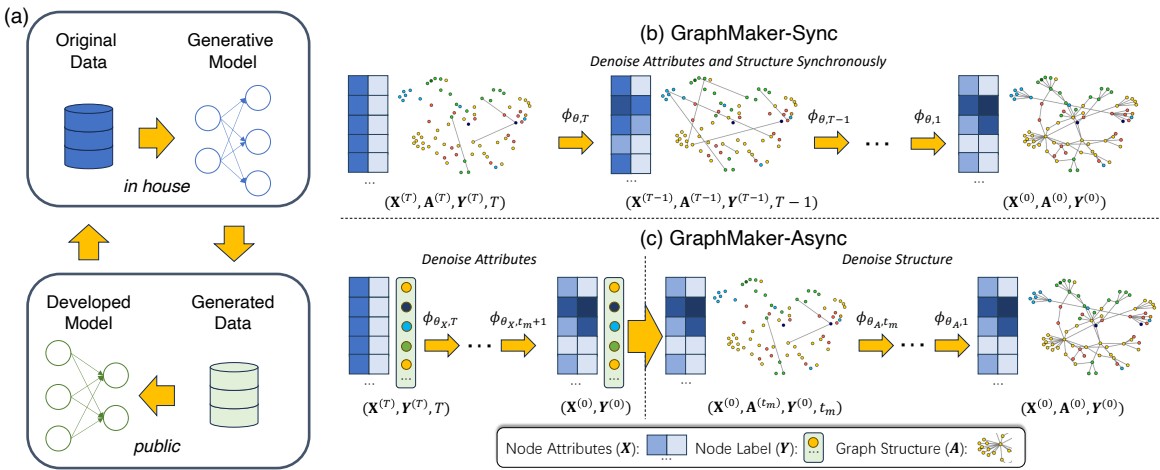

Figure 1: (a) Data generation for public usage. (b) & (c): Generation process with two GraphMaker variants.

approach is especially beneficial in fields like social/financial network studies, where synthetic versions can be used for developing and benchmarking graph learning models, avoiding the need to access the original data. Most of such scenarios require generating large attributed graphs. Fig. 1 (a) illustrates this procedure.

Traditional approaches primarily use statistical random graph models to generate graphs (Erdős & Rényi, 1959; Holland et al., 1983; Barabási & Albert, 2002). These models often contain very few parameters such as triangle numbers, edge densities, and degree distributions, which often suffer from limited model capacity. AGM (Pfeiffer et al., 2014) extends these models for additionally generating node attributes, but it can only handle very few attributes due to the curse of dimensionality. Recent research on deep generative models of graphs has introduced more expressive data-driven methods. Most of those works solely focus on graph structure generation (Kipf & Welling, 2016; You et al., 2018b; Bojchevski et al., 2018; Li et al., 2018; Liao et al., 2019). Other works study molecular graph generation that involves attributes, but these graphs are often small-scale, and only consist of tens of nodes per graph and a single categorical attribute per node (Jin et al., 2018; Liu et al., 2018; You et al., 2018a; De Cao & Kipf, 2018).

Diffusion models have achieved remarkable success in image generation (Ho et al., 2020; Rombach et al., 2022) by learning a model to denoise a noisy sample. They have been recently extended to graph structure and molecule generation. Niu et al. (2020) corrupts real graphs by adding Gaussian noise to all entries of dense adjacency matrices. GDSS (Jo et al., 2022) extends the idea for molecule generation. DiGress (Vignac et al., 2023) addresses the discretization challenge due to Gaussian noise. Specifically, DiGress employs D3PM (Austin et al., 2021) to edit individual categorical node and edge types. However, DiGress is still designed for small molecule generation. EDGE (Chen et al., 2023) and GRAPHARM (Kong et al., 2023) propose autoregressive diffusion models for graph structure and molecule generation. Overall, all previous graph diffusion models are not designed to generate large attributed graphs and may suffer from various limitations for this purpose.

Developing diffusion models of large-attributed graphs is challenging on several aspects. First, a large attributed graph presents substantially different patterns from molecular graphs, exhibiting complex correlations between high-dimensional node attributes and graph structure. Second, generating large graphs poses challenges to model scalability as the number of possible edges exhibits quadratic growth in the number of nodes, let alone the potential hundreds or thousands of attributes per node. Third, how to evaluate the quality of the generated graphs remains an open problem. Although deep models have the potential to capture more complicated data distribution and correlations, most previous studies just evaluate high-level statistics characterizing structural properties such as node degree distributions, and clustering coefficients of the generated graphs, which can be already captured well by early-day statistical models (Pfeiffer et al., 2012; Seshadri et al., 2012). Therefore, a finer-grained way to evaluate the generated large-attributed

graphs is needed for understanding the strengths and limitations of deep generative models, in particular, if the generated synthetic graphs are to be further applied to develop and benchmark graph learning models.

Here we present GraphMaker, a diffusion model for generating large attributed graphs. First, we observe that denoising graph structure and node attributes simultaneously with a diffusion model (Fig. 1 (b)) may yield poor performance in capturing the patterns of either component and their joint patterns for large attributed graphs, which is different from the existing observations for molecule generation (Jo et al., 2022). Instead, we propose a generalized diffusion process that allows asynchronous corruption and denoising of the two components (Fig. 1 (c)), which shows better performance in learning the generative models for many real-world graphs. Second, for scalability challenges, we employ a minibatch strategy for structure prediction to avoid enumerating all node pairs per gradient update during training, and design a new message-passing neural network (MPNN) to efficiently encode the data. Third, motivated by the end goal of generating synthetic graphs that preserve utility for developing graph learning models (Fig. 1 (a)), we propose to utilize ML models, including graph neural networks (GNNs) (Kipf & Welling, 2016; 2017; Wu et al., 2019; Gasteiger et al., 2019), by training them on the generated graphs and subsequently evaluating their performance on the original graphs. When equipped with strong discriminative models, our evaluation protocol allows more fine-grained examination of attribute-structure correlation, which is complementary to the evaluation based on high-level graph statistics. For the scenarios targeting the task of node classification, each node is additionally associated with a label. In this case, GraphMaker can perform conditional generation given node labels, which further improves the data utility for ML model development.

Extensive studies on real-world networks with up to more than $13K$ nodes, $490K$ edges, and $1K$ attributes demonstrate that GraphMaker overall significantly outperforms the baselines in producing graphs with realistic properties and high utility for graph ML model development. For evaluation on graph ML tasks, GraphMaker achieves the best performance for 80% of cases across all datasets. For property evaluation, GraphMaker achieves the best performance for 50% of cases. In addition, we demonstrate the capability of GraphMaker in generating diverse novel graphs.

## 2 GraphMaker

### 2.1 Preliminaries and problem formulation

Consider an undirected graph $G = (\mathcal{V}, \mathcal{E}, \mathbf{X})$ with the node set $\mathcal{V}$ and the edge set $\mathcal{E}$. Let $N = |\mathcal{V}|$ denote the number of nodes. Each node $v \in \mathcal{V}$ is associated with $F$-dimensional categorical attributes $\mathbf{X}_v$. Note that we are to generate graphs without edge attributes to show a proof of concept, while the method can be extended for the case with edge attributes. We aim to learn $\mathbb{P}_{\mathcal{G}}$ with $\mathbb{P}_{\mathcal{G}}^\theta$ based on one graph $G$ sampled from it, which can be used to generate synthetic graphs for further graph learning model development. In Section 2.6, we provide some reasoning on why the distribution may be possibly learned from one single observation. In the scenarios targeting node classification tasks, each node $v$ is additionally associated with a label $\mathbf{Y}_v$, which require learning $\mathbb{P}_{\mathcal{G} \times \mathcal{Y}}$ from $(G, \mathbf{Y})$ instead.

### 2.2 Forward diffusion process and reverse process

Our diffusion model extends D3PM (Austin et al., 2021) to large attributed graphs. There are two phases. The forward diffusion process corrupts the raw data by progressively perturbing its node attributes or edges. Let $\mathbf{X} \in \mathbb{R}^{N \times F \times C_X}$ be the one-hot encoding of the categorical node attributes, where for simplicity we assume all attributes to have $C_X$ possible classes. By treating the absence of edge as an edge type, we denote the one-hot encoding of all edges by $\mathbf{A} \in \mathbb{R}^{N \times N \times 2}$. Let $G^{(0)}$ be the real graph data $(\mathbf{X}, \mathbf{A})$.

We individually corrupt node attributes and edges of $G^{(t-1)}$ into $G^{(t)}$ by first obtaining noisy distributions with transition matrices and then sampling from them. By composing the noise over multiple time steps, for $G^{(t)} = (\mathbf{X}^{(t)}, \mathbf{A}^{(t)})$ at time step $t \in [T]$, we have $q(\mathbf{X}_{v,f}^{(t)}|\mathbf{X}_{v,f}) = \mathbf{X}_{v,f}\bar{\mathbf{Q}}_{X_f}^{(t)}$ for any $v \in [N]$, $f \in [F]$ and $q(\mathbf{A}^{(t)}|\mathbf{A}) = \mathbf{A}\bar{\mathbf{Q}}_A^{(t)}$. To ensure $G^{(t)}$ to be undirected, $\bar{\mathbf{Q}}_A^{(t)}$ is only applied to the upper triangular part of $\mathbf{A}$. The adjacency matrix of $G^{(t)}$, i.e., $\mathbf{A}^{(t)}$, is then constructed by symmetrizing the matrix after sampling. We consider a general formulation of $\bar{\mathbf{Q}}_{X_f}^{(t)}$ and $\bar{\mathbf{Q}}_A^{(t)}$ that allows *asynchronous* corruption of node

attributes and edges. Let $\mathcal{T}_X = \{t_X^1, \cdots, t_X^{T_X}\} \subset [T]$ be the time steps of corrupting node attributes, and $\mathcal{T}_A = \{t_A^1, \cdots, t_A^{T_A}\} \subset [T]$ be the time steps of corrupting edges. Then,

$$\bar{\mathbf{Q}}_{X_f}^{(t)} = \bar{\alpha}_{\gamma_X(t)}\mathbf{I} + \left(1 - \bar{\alpha}_{\gamma_X(t)}\right)\mathbf{1}\mathbf{m}_{X_f}^\top \tag{1}$$

$$\bar{\mathbf{Q}}_A^{(t)} = \bar{\alpha}_{\gamma_A(t)}\mathbf{I} + \left(1 - \bar{\alpha}_{\gamma_A(t)}\right)\mathbf{1}\mathbf{m}_A^\top \tag{2}$$

where $\bar{\alpha}_{\gamma_Z(t)} = \bar{\alpha}_{|\{t_Z^i | t_Z^i \le t\}|}$ if $t \le t_Z^{T_Z}$ or otherwise $\bar{\alpha}_{\gamma_Z(t)} = \bar{\alpha}_0 = 1$, for $Z \in \{X, A\}$. We consider the popular cosine schedule in this paper (Nichol & Dhariwal, 2021), where $\bar{\alpha}_{\gamma_Z(t)} = \cos^2(\frac{\pi}{2}\frac{\gamma_Z(t)/|\mathcal{T}_Z|+s}{1+s})$ for some small $s$, but the model works for other schedules as well. $\mathbf{I}$ is the identity matrix, $\mathbf{1}$ is a one-valued vector, $\mathbf{m}_{X_f}$ is the empirical marginal distribution of the $f$-th node attribute in the real graph, $\mathbf{m}_A$ is the empirical marginal distribution of the edge existence in the real graph, and $^\top$ denotes transpose.

During the reverse process, the second phase of the diffusion model, a denoising network $\phi_{\theta,t}$ is trained to perform one-step denoising $p_\theta(G^{(t-1)}|G^{(t)}, t)$. Once trained, we can iteratively apply this denoising network to a noisy graph sampled from the prior distribution $\prod_{v=1}^{\hat{N}} \prod_{f=1}^F \mathbf{m}_{X_f} \prod_{1 \le u \le \hat{N}} \prod_{u < v \le \hat{N}} \mathbf{m}_A$ for data generation. While we consider $\hat{N} = N$ in this paper, the model is capable of generating graphs of a different size. We model $p_\theta(G^{(t-1)}|G^{(t)}, t)$ as a product of conditionally independent distributions over node attributes and edges.

$$p_\theta(G^{(t-1)}|G^{(t)}, t) = \prod_{v=1}^N \prod_{f=1}^F p_\theta(\mathbf{X}_{v,f}^{(t-1)}|G^{(t)}, t)$$
$$\prod_{1 \le u < v \le N} p_\theta(\mathbf{A}_{u,v}^{(t-1)}|G^{(t)}, t) \tag{3}$$

Sohl-Dickstein et al. (2015) and Song & Ermon (2019) show that we can train the denoising network to predict $G^{(0)}$ instead of $G^{(t-1)}$ as long as $\int q(G^{(t-1)}|G^{(t)}, t, G^{(0)})dp_\theta(G^{(0)}|G^{(t)}, t)$ has a closed-form expression. This holds for our case. By the Bayes rule, we have $q(\mathbf{X}_{v,f}^{(t-1)}|G^{(t)}, t, G^{(0)}) \propto \mathbf{X}_{v,f}^{(t)}(\mathbf{Q}_{X_f}^{(t)})^\top \odot \mathbf{X}_{v,f}\bar{\mathbf{Q}}_{X_f}^{(t-1)}$, where $\mathbf{Q}_{X_f}^{(t)} = (\bar{\mathbf{Q}}_{X_f}^{(t-1)})^{-1}\bar{\mathbf{Q}}_{X_f}^{(t)}$, and $\odot$ is the elementwise product. Similarly, we have $q(\mathbf{A}_{u,v}^{(t-1)}|G^{(t)}, t, G^{(0)}) \propto \mathbf{A}_{u,v}^{(t)}(\mathbf{Q}_A^{(t)})^\top \odot \mathbf{A}_{u,v}\bar{\mathbf{Q}}_A^{(t-1)}$.

## 2.3 Two instantiations

To model the complex correlations between node attributes and graph structure, we study two particular instantiations of GraphMaker, named GraphMaker-Sync and GraphMaker-Async. When categorical node label $\mathbf{Y}$ is present, as in the task of node classification, we can treat it as an extra node attribute. For simplicity, we do not distinguish it from $\mathbf{X}$ for the rest of this sub-section unless stated explicitly.

**GraphMaker-Sync (GMaker-S).** GraphMaker-Sync employs a forward diffusion process that simultaneously corrupts node attributes and edges for all time steps, which corresponds to setting $\mathcal{T}_X = \mathcal{T}_A = [T]$. The denoising network is trained to recover clean node attributes and edges from corrupted node attributes and edges. During generation, it first samples noisy attributes $\mathbf{X}^{(T)}$, and noisy edges $\mathbf{A}^{(T)}$ from the prior distributions. It then repeatedly invokes the denoising network $\phi_{\theta,t}$ to predict $G^{(0)}$ for computing the posterior distribution $p_\theta(G^{(t-1)}|G^{(t)}, t)$, and then samples $G^{(t-1)}$. Fig. 1 (b) provides an illustration.

**GraphMaker-Async (GMaker-A).** In practice, we find that GraphMaker-Sync cannot properly capture the correlations between high-dimensional node attributes, graph structure, and node label, as shown in later Sections 3.2 and 3.3. Previous work has pointed out that a failure to model edge dependency in graph structure generation yields poor generation quality in capturing patterns like triangle count (Chanpuriya et al., 2021). We suspect that this problem stems from synchronous refinement of node attributes and graph structure, hence we propose to denoise node attributes and graph structure asynchronously instead. We consider a simple and practically effective configuration as a proof of concept, which partitions $[1, T]$ into two subintervals $\mathcal{T}_A = [1, t_m]$ and $\mathcal{T}_X = [t_m + 1, T]$. The denoising network $\phi_{\theta,t}$ consists of two sub-networks. $\phi_{\theta_X,t}$ is an MLP trained to reconstruct node attributes $\mathbf{X}^{(0)}$ given $(\mathbf{X}^{(t)}, t)$. $\phi_{\theta_A,t}$ is trained to reconstruct

edges given $(\mathbf{A}^{(t)}, \mathbf{X}^{(0)}, t)$. During generation, it first samples noisy attributes $\mathbf{X}^{(T)}$, then repeatedly invokes $\phi_{\theta_X, t}$ until the generation of node attributes is finished. Finally, it samples noisy edges $\mathbf{A}^{(T)}$ and invokes $\phi_{\theta_A, t}(\mathbf{A}^{(t)}, \mathbf{X}^{(0)}, t)$ repeatedly to complete the edge generation. Fig. 1 (c) provides a visual illustration.

## 2.4 Conditional generation given node labels

An important learning task for large attributed graphs is node classification, where each node is associated with a label. To generate graph data with node labels, one naïve way is to simply generate node labels as extra node attributes. While this idea is straightforward, it may not perform the best at capturing the correlation between node label and other components, as empirically demonstrated in Section 3.2 and 3.3. To further improve the generation quality for developing or benchmarking node classification models, we propose conditional graph generation. Conditional generative models have been proven valuable in many applications, such as generating images consistent with a target text description (Rombach et al., 2022) or material candidates that satisfy desired properties (Vignac et al., 2023), where controllable generation is often desirable.

Specifically, since the node label distribution $\mathbb{P}_\mathcal{Y}$ can be estimated from the real node labels, we can simplify the problem by learning the conditional distribution $\mathbb{P}_{\mathcal{G}|\mathcal{Y}}$ rather than $\mathbb{P}_{\mathcal{G} \times \mathcal{Y}}$, which essentially requires learning a conditional generative model. This corresponds to setting $\mathbf{Y}^{(0)} = \cdots = \mathbf{Y}^{(T)}$ in figure 1 (b) & (c). During training, the denoising network is supplied with uncorrupted node labels. During generation, node labels are sampled and fixed at the beginning to guide the generation of the rest node attributes and graph structure. We notice that conditional generation gives a further boost in the accuracy for developing node classification models through the generated graphs.

## 2.5 Scalable denoising network architecture

Large attributed graphs consist of more than thousands of nodes. This poses severe challenges to the scalability of the denoising network. We address the challenge with both the encoder and the decoder of the denoising network. The encoder computes representations of $G^{(t)}$ and the decoder makes predictions of node attributes and edges with them.

**Graph encoder.** To enhance the scalability of the graph encoder, we propose a message passing neural network (MPNN) with complexity $O(|\mathcal{E}|)$ instead of using a graph transformer (Dwivedi & Bresson, 2021) with complexity $O(N^2)$, which was employed by previous graph diffusion generative models (Vignac et al., 2023). Empirically, we find that with a 16-GB GPU, we can use at most a single graph transformer layer, with four attention heads and a hidden size of 16 per attention head, for a graph with approximately two thousand nodes and one thousand attributes. Furthermore, while in theory graph transformers surpass MPNNs in modeling long-range dependencies owing to non-local interactions, it is still debatable whether long-range dependencies are really needed for large-attributed graphs. In particular, graph transformers have not demonstrated superior performance on standard benchmarks for large attributed graphs like OGB (Hu et al., 2020).

Let $\mathbf{A}^{(t)}$ and $\mathbf{X}^{(t)}$ respectively be the one-hot encoding of the edges and node attributes for the time step $t$. The encoder first uses a multilayer perceptron (MLP) to transform $\mathbf{X}^{(t)}$ and the time step $t$ into hidden representations $\mathbf{X}^{(t,0)}$ and $\mathbf{h}^{(t)}$. It then employs multiple MPNN layers to update attribute representations. For any node $v \in \mathcal{V}$, $\mathbf{X}_v^{(t,l+1)} = \sigma(\mathbf{W}_{T \rightarrow X}^{(l)} \mathbf{h}^{(t)} + \mathbf{b}_X^{(l)} + \sum_{u \in \mathcal{N}^{(t)}(v)} \frac{1}{|\mathcal{N}^{(t)}(v)|} \mathbf{X}_u^{(t,l)} \mathbf{W}_{X \rightarrow X}^{(l)})$, where $\mathbf{W}_{T \rightarrow X}^{(l)}, \mathbf{W}_{X \rightarrow X}^{(l)}$, and $\mathbf{b}_X^{(l)}$ are learnable matrices and vectors. $\mathcal{N}^{(t)}(v)$ consists of $v$ and the neighbors of $v$ corresponding to $\mathbf{A}^{(t)}$. $\sigma$ consists of a ReLU layer Jarrett et al. (2009), a LayerNorm layer Ba et al. (2016), and a dropout layer Srivastava et al. (2014). The encoder computes the final node representations with $\mathbf{H}_v^{(t)} = \mathbf{X}_v^{(t,0)} \| \mathbf{X}_v^{(t,1)} \| \cdots \| \mathbf{h}^{(t)}$, which combines the representations from all MPNN layers as in JK-Nets Xu et al. (2018). We employ two separate encoders for node attribute prediction and edge prediction.

For conditional generation as discussed in Section 2.4, we treat node labels in a way different from other node attributes. Specifically, we update node label representations separately $\mathbf{Y}_v^{(0,l+1)} = \sigma(\mathbf{b}_Y^{(l)} + \sum_{u \in \mathcal{N}^{(t)}(v)} \frac{1}{|\mathcal{N}^{(t)}(v)|} \mathbf{Y}_u^{(0,l)} \mathbf{W}_{Y \rightarrow Y}^{(l)})$, where $\mathbf{W}_{Y \rightarrow Y}^{(l)}$ and $\mathbf{b}_Y^{(l)}$ are learnable. Note that here we do not cor-

rupt node labels, so there is no need to incorporate the time-step encoding $\mathbf{h}^{(t)}$. The label representations of layer $l$, $\{\mathbf{Y}_u^{(0,l)}\}_{u \in \mathcal{N}^{(t)}(v)}$, are used in computing $\mathbf{X}_v^{(t,l+1)}$ with $\mathbf{X}_v^{(t,l+1)} = \sigma(\mathbf{W}_{T \to X}^{(l)} \mathbf{h}^{(t)} + \mathbf{b}_X^{(l)} + \sum_{u \in \mathcal{N}^{(t)}(v)} \frac{1}{|\mathcal{N}^{(t)}(v)|}[\mathbf{X}_u^{(t,l)} \| \mathbf{Y}_u^{(0,l)}]\mathbf{W}_{X \to X}^{(l)})$ and $\mathbf{H}_v^{(t)}$ with $\mathbf{H}_v^{(t)} = \mathbf{X}_v^{(t,0)} \| \mathbf{X}_v^{(t,1)} \| \cdots \| \mathbf{Y}_v^{(0,0)} \| \mathbf{Y}_v^{(0,1)} \| \cdots \| \mathbf{h}^{(t)}$.

**Decoder.** The decoder performs node attribute prediction from $\mathbf{H}_v^{(t)}$ in the form of multi-label node classification. To predict edge existence between nodes $u, v \in \mathcal{V}$ in an undirected graph, the decoder performs binary node pair classification. It first computes the elementwise product $\mathbf{H}_u^{(t)} \odot \mathbf{H}_v^{(t)}$, and then applies an MLP to transform this product into the final score. Due to limited GPU memory, it is intractable to perform edge prediction for all $N^2$ node pairs at once. During training, we randomly choose a subset of the node pairs for a gradient update. For graph generation, we first compute the representations of all nodes and then perform edge prediction over minibatches of node pairs.

### 2.6 Can a graph generative model be learned from a single graph?

It remains to be answered why we can learn the graph distribution from only one sampled graph. Due to the permutation equivariance inherent in our MPNN architecture, the derived probability distribution is permutation invariant. In other words, altering the node ordering in a graph doesn't affect the model's likelihood of generating it. This suggests that our model is primarily focused on identifying common patterns across nodes instead of focusing on the unique details of individual nodes. In this case, a single, expansive graph can offer abundant learning examples—each node centered subgraph acts as a distinct data point.

This raises an intriguing query – should a model for large graph generation capture individual node characteristics? We think the answer depends on the use cases. If the goal is to analyze population-level trends, individual node details will be distractions. Note that traditional graph generative models like ER (Erdős & Rényi, 1959), and its degree-corrected counterpart (Seshadhri et al., 2012) only capture population-level statistics like degree distributions. For scenarios that involve sharing synthetic graphs with privacy considerations, omitting the node-specific details is advantageous. Conversely, if node-specific analysis is essential, our model might fall short. Overall, there's always a tradeoff between a model's capabilities, the data at hand, the desired detail level, and privacy considerations. We think of our work as a first step and will let future works dive deeper into this issue. In Section 3.5, we empirically study this point. Indeed, adopting a more expressive model with node positional encodings (Eliasof et al., 2023) that may capture some individual node details does not consistently improve the quality of the generated graphs in our evaluation.

## 3 Experiments

In this section, we aim to answer the following questions with empirical studies. **Q1)** For the end goal of using synthetic graphs to develop and benchmark ML models, how does GraphMaker perform against the existing generative models in preserving data utility for ML? **Q2)** For the studies of network science, how does GraphMaker perform against the existing generative models in capturing structural properties? **Q3)** For the potential application of privacy-preserving synthetic data sharing, generative models should avoid merely reproducing the original graph. To what extent do the generated synthetic graphs replicate the original graph? **Q4)** Does capturing individual-level node characteristics benefit capturing population-level patterns?

### 3.1 Datasets and baselines

**Datasets:** We utilize three large attributed networks for evaluation. Cora is a citation network depicting citation relationships among papers (Sen et al., 2008), with binary node attributes indicating the presence of keywords and node labels representing paper categories. Amazon Photo and Amazon Computer are product co-purchase networks, where two products are connected if they are frequently purchased together (Shchur et al., 2018). The node attributes indicate the presence of words in product reviews and node labels represent product categories. See Appendix A.1 for the dataset statistics. Notably, Amazon Computer (13K nodes, 490K edges) is an order of magnitude larger than graphs adopted for evaluation by previous deep generative

models of graph structures (Chen et al., 2023; Kong et al., 2023), and hence it is a challenging testbed for model scalability.

**Baselines:** To understand the strengths and weaknesses of the previous diffusion models in large attributed graph generation, we consider two state-of-the-art diffusion models developed for graph generation, GDSS (Jo et al., 2022) and EDGE (Chen et al., 2023). For non-diffusion deep learning (DL) methods, we adopt feature-based matrix factorization (FMF) (Chen et al., 2012), graph auto-encoder (GAE), and variational graph auto-encoder (VGAE) (Kipf & Welling, 2016). In addition, we compare against the Erdős–Rényi (ER) model (Erdős & Rényi, 1959), a traditional statistical random graph model. None of the non-diffusion baselines inherently possesses the capability for node attribute generation. Therefore, we equip them with the empirical marginal distribution $p(\mathbf{Y}) \prod_v \prod_f p(\mathbf{X}_{v,f}|\mathbf{Y}_v)$ for attribute and label generation. Node attributes and label generated this way yield competitive classification performance, as shown in Appendix A.3. For more details on baseline experiments, see Appendix A.2.

### 3.2 Evaluation for ML utility (Q1)

To evaluate the utility of the generated graphs for developing ML models, we introduce a novel evaluation protocol based on discriminative models. We train one model on the training set of the original graph $(G, \mathbf{Y})$ and another model on the training set of a generated graph $(\hat{G}, \hat{\mathbf{Y}})$. We then evaluate the two models on the test set of the original graph to obtain two performance metrics $\mathrm{ACC}(G|G)$ and $\mathrm{ACC}(G|\hat{G})$. If the ratio $\mathrm{ACC}(G|\hat{G})/\mathrm{ACC}(G|G)$ is close to one, then the generated graph is considered as having a utility similar to the original graph for training the particular model. We properly tune each model to ensure a fair comparison, see Appendix A.3 for more details.

**Node classification.** Node classification models evaluate the correlations between unlabeled graphs and their corresponding node labels. Given our focus on the scenario with a single large graph, we approach the semi-supervised node classification problem. We randomly split a generated dataset so that the number of labeled nodes in each class and each subset is the same as that in the original dataset. For discriminative models, we choose three representative GNNs – SGC (Wu et al., 2019), GCN (Kipf & Welling, 2017), and APPNP (Gasteiger et al., 2019). As they employ different orders for message passing and prediction, this combination allows for examining data patterns more comprehensively. To scrutinize the retention of higher-order patterns, we employ two variants for each model, one with a single message passing layer denoted by $1-*$ and another with multiple message passing layers denoted by $L-*$. In addition, we directly evaluate the correlations between node attributes and node label with MLP in appendix A.3.

**Link prediction.** This task requires predicting missing edges in an incomplete graph, potentially utilizing node attributes and node labels. To prevent label leakage from dataset splitting, we split the edges corresponding to the upper triangular adjacency matrix into different subsets and then add reverse edges after the split. We consider two types of discriminative models – Common Neighbor (CN) (Liben-Nowell & Kleinberg, 2003) and GAE (Kipf & Welling, 2016). CN is a traditional method that predicts edge existence if the number of common neighbors shared by two nodes exceeds a threshold. GAE is an MPNN-based model that integrates the information of graph structure, node attributes, and node labels. As before, we consider two variants for GAE. Following the practice of Kipf & Welling (2016), we adopt ROC-AUC as the evaluation metric.

Table 1 presents the results. Unless otherwise mentioned, we report all results in this paper based on three different random seeds. For the performance of the discriminative models trained and evaluated on the original graph, see Appendix A.3. Out of 24 cases, GraphMaker variants perform better or no worse than the baselines for 20 of them ($\approx 83\%$).

Diffusion models that asynchronously denoise node attributes and graph structure (GraphMaker-Async (GMaker-A) and EDGE) outperform the synchronous ones (GraphMaker-Sync (GMaker-S) and GDSS) by a large margin. This phenomenon is in contrast to the previous observation made for molecule generation by the GDSS paper (Jo et al., 2022), reflecting the drastically different patterns between a large attributed graph and molecular graphs. EDGE proposes a degree guidance mechanism, where the graph generation is conditioned on node degrees pre-determined from the empirical node degree distribution. While it achieves competitive performance on Cora, it falls short on Amazon Photo, which might be due to degree constraints.

Table 1: Evaluation with discriminative models. Best results are in **bold**. OOM stands for out of memory.

| | Cora | | | | | | | |
|---|---|---|---|---|---|---|---|---|
| Model | Node Classification → 1 | | | | | Link Prediction → 1 | | |
| | 1-SGC | L-SGC | L-GCN | 1-APPNP | L-APPNP | CN | 1-GAE | L-GAE |
| ER | $0.55 \pm 0.08$ | $0.65 \pm 0.17$ | $0.64 \pm 0.22$ | $0.83 \pm 0.02$ | $0.87 \pm 0.04$ | $0.80 \pm 0.13$ | $0.90 \pm 0.07$ | $0.79 \pm 0.05$ |
| FMF | $1.06 \pm 0.01$ | $0.83 \pm 0.12$ | $0.79 \pm 0.16$ | $1.04 \pm 0.02$ | $1.01 \pm 0.02$ | $0.71 \pm 0.00$ | $0.95 \pm 0.01$ | $0.74 \pm 0.03$ |
| GAE | $1.06 \pm 0.01$ | $0.85 \pm 0.19$ | $0.76 \pm 0.17$ | $1.04 \pm 0.02$ | $1.01 \pm 0.02$ | $0.71 \pm 0.00$ | $0.96 \pm 0.01$ | $0.75 \pm 0.07$ |
| VGAE | $\mathbf{1.00 \pm 0.01}$ | $0.70 \pm 0.23$ | $0.51 \pm 0.16$ | $1.04 \pm 0.02$ | $\mathbf{1.00 \pm 0.02}$ | $0.71 \pm 0.00$ | $0.97 \pm 0.00$ | $0.71 \pm 0.03$ |
| GDSS | $0.28 \pm 0.10$ | $0.25 \pm 0.07$ | $0.26 \pm 0.08$ | $0.23 \pm 0.04$ | $0.23 \pm 0.06$ | $0.88 \pm 0.14$ | $0.82 \pm 0.07$ | $0.74 \pm 0.05$ |
| EDGE | $0.92 \pm 0.03$ | $1.01 \pm 0.01$ | $\mathbf{1.00 \pm 0.02}$ | $0.96 \pm 0.02$ | $\mathbf{1.00 \pm 0.02}$ | $\mathbf{1.00 \pm 0.00}$ | $0.97 \pm 0.00$ | $0.97 \pm 0.01$ |
| GMaker-S | $0.40 \pm 0.06$ | $0.41 \pm 0.07$ | $0.40 \pm 0.00$ | $0.41 \pm 0.00$ | $0.40 \pm 0.02$ | $0.88 \pm 0.14$ | $0.92 \pm 0.06$ | $0.83 \pm 0.05$ |
| GMaker-A | $0.88 \pm 0.04$ | $0.97 \pm 0.04$ | $0.99 \pm 0.03$ | $0.93 \pm 0.03$ | $\mathbf{1.00 \pm 0.02}$ | $\mathbf{1.00 \pm 0.00}$ | $\mathbf{0.98 \pm 0.00}$ | $\mathbf{0.99 \pm 0.00}$ |
| GMaker-S (cond.) | $0.93 \pm 0.02$ | $1.02 \pm 0.02$ | $1.01 \pm 0.02$ | $\mathbf{0.98 \pm 0.01}$ | $1.01 \pm 0.02$ | $\mathbf{1.00 \pm 0.00}$ | $\mathbf{0.98 \pm 0.00}$ | $0.98 \pm 0.01$ |
| GMaker-A (cond.) | $0.95 \pm 0.02$ | $\mathbf{1.00 \pm 0.02}$ | $\mathbf{1.00 \pm 0.03}$ | $0.96 \pm 0.03$ | $1.01 \pm 0.02$ | $\mathbf{1.00 \pm 0.00}$ | $\mathbf{0.98 \pm 0.00}$ | $\mathbf{0.99 \pm 0.00}$ |

| | Amazon Photo | | | | | | | |
|---|---|---|---|---|---|---|---|---|
| Model | Node Classification → 1 | | | | | Link Prediction → 1 | | |
| | 1-SGC | L-SGC | L-GCN | 1-APPNP | L-APPNP | CN | 1-GAE | L-GAE |
| ER | $0.38 \pm 0.10$ | $0.45 \pm 0.17$ | $0.16 \pm 0.12$ | $0.79 \pm 0.04$ | $0.80 \pm 0.04$ | $0.85 \pm 0.22$ | $0.95 \pm 0.01$ | $0.63 \pm 0.15$ |
| FMF | $0.84 \pm 0.09$ | $0.37 \pm 0.17$ | $0.12 \pm 0.01$ | $0.85 \pm 0.03$ | $0.72 \pm 0.05$ | $0.54 \pm 0.00$ | $0.99 \pm 0.00$ | $0.82 \pm 0.01$ |
| GAE | $0.60 \pm 0.10$ | $0.18 \pm 0.03$ | $0.07 \pm 0.04$ | $0.84 \pm 0.03$ | $0.72 \pm 0.08$ | $0.54 \pm 0.00$ | $0.99 \pm 0.00$ | $0.74 \pm 0.03$ |
| VGAE | $0.89 \pm 0.11$ | $0.42 \pm 0.08$ | $0.19 \pm 0.09$ | $0.78 \pm 0.02$ | $0.81 \pm 0.00$ | $0.54 \pm 0.00$ | $0.99 \pm 0.00$ | $0.79 \pm 0.02$ |
| GDSS | OOM | OOM | OOM | OOM | OOM | OOM | OOM | OOM |
| EDGE | $0.78 \pm 0.08$ | $0.76 \pm 0.09$ | $0.76 \pm 0.05$ | $0.90 \pm 0.02$ | $0.92 \pm 0.02$ | $\mathbf{1.00 \pm 0.00}$ | $0.97 \pm 0.01$ | $0.91 \pm 0.03$ |
| GMaker-S | $0.12 \pm 0.03$ | $0.19 \pm 0.08$ | $0.11 \pm 0.05$ | $0.18 \pm 0.06$ | $0.21 \pm 0.05$ | $0.54 \pm 0.00$ | $0.85 \pm 0.01$ | $0.84 \pm 0.00$ |
| GMaker-A | $0.97 \pm 0.06$ | $0.98 \pm 0.06$ | $0.93 \pm 0.03$ | $\mathbf{0.93 \pm 0.01}$ | $0.92 \pm 0.02$ | $\mathbf{1.00 \pm 0.00}$ | $\mathbf{1.00 \pm 0.00}$ | $0.98 \pm 0.01$ |
| GMaker-S (cond.) | $\mathbf{1.00 \pm 0.06}$ | $\mathbf{1.00 \pm 0.06}$ | $0.89 \pm 0.01$ | $0.89 \pm 0.01$ | $0.89 \pm 0.02$ | $0.99 \pm 0.00$ | $0.98 \pm 0.00$ | $0.98 \pm 0.00$ |
| GMaker-A (cond.) | $1.12 \pm 0.11$ | $1.02 \pm 0.05$ | $\mathbf{0.94 \pm 0.00}$ | $\mathbf{0.93 \pm 0.03}$ | $\mathbf{0.94 \pm 0.01}$ | $\mathbf{1.00 \pm 0.00}$ | $\mathbf{1.00 \pm 0.00}$ | $\mathbf{0.99 \pm 0.00}$ |

| | Amazon Computer | | | | | | | |
|---|---|---|---|---|---|---|---|---|
| Model | Node Classification → 1 | | | | | Link Prediction → 1 | | |
| | 1-SGC | L-SGC | L-GCN | 1-APPNP | L-APPNP | CN | 1-GAE | L-GAE |
| ER | $0.28 \pm 0.20$ | $0.38 \pm 0.11$ | $0.16 \pm 0.10$ | $0.76 \pm 0.15$ | $0.87 \pm 0.02$ | $0.82 \pm 0.20$ | $0.96 \pm 0.03$ | $0.88 \pm 0.03$ |
| FMF | $0.20 \pm 0.06$ | $0.11 \pm 0.08$ | $0.11 \pm 0.03$ | $0.91 \pm 0.03$ | $0.71 \pm 0.26$ | $0.54 \pm 0.00$ | $\underline{0.99 \pm 0.00}$ | $0.83 \pm 0.02$ |
| GAE | $0.43 \pm 0.13$ | $0.29 \pm 0.04$ | $0.35 \pm 0.19$ | $0.92 \pm 0.02$ | $0.79 \pm 0.03$ | $0.54 \pm 0.00$ | $0.98 \pm 0.00$ | $0.87 \pm 0.07$ |
| VGAE | $0.34 \pm 0.05$ | $0.37 \pm 0.01$ | $0.12 \pm 0.07$ | $0.80 \pm 0.15$ | $0.62 \pm 0.17$ | $0.54 \pm 0.00$ | $0.97 \pm 0.02$ | $0.87 \pm 0.03$ |
| GDSS | OOM | OOM | OOM | OOM | OOM | OOM | OOM | OOM |
| EDGE | OOM | OOM | OOM | OOM | OOM | OOM | OOM | OOM |
| GMaker-S | $0.32 \pm 0.19$ | $0.32 \pm 0.11$ | $0.07 \pm 0.03$ | $0.18 \pm 0.04$ | $0.09 \pm 0.05$ | $\mathbf{1.00 \pm 0.00}$ | $0.84 \pm 0.01$ | $0.77 \pm 0.08$ |
| GMaker-A | $0.92 \pm 0.07$ | $0.86 \pm 0.15$ | $0.94 \pm 0.01$ | $0.95 \pm 0.01$ | $0.97 \pm 0.02$ | $\mathbf{1.00 \pm 0.00}$ | $0.96 \pm 0.00$ | $0.98 \pm 0.01$ |
| GMaker-S (cond.) | $0.76 \pm 0.04$ | $0.87 \pm 0.04$ | $0.92 \pm 0.00$ | $0.89 \pm 0.02$ | $0.92 \pm 0.01$ | $0.85 \pm 0.01$ | $0.97 \pm 0.01$ | $0.98 \pm 0.03$ |
| GMaker-A (cond.) | $\mathbf{1.00 \pm 0.18}$ | $\mathbf{1.00 \pm 0.10}$ | $\mathbf{0.96 \pm 0.02}$ | $\mathbf{0.96 \pm 0.02}$ | $\mathbf{0.98 \pm 0.03}$ | $0.82 \pm 0.01$ | $0.97 \pm 0.00$ | $\mathbf{1.01 \pm 0.00}$ |

Moreover, conditioning on degree distribution reduces the novelty of the generated graphs. In terms of scalability, GraphMaker is the only diffusion model that does not encounter the out-of-memory (OOM) error on a 48-GB GPU. As GDSS denoises dense adjacency matrices, it has a complexity of $O(N^2)$ for representation computation even with an MPNN. Therefore, it has a worse scalability than EDGE. Overall, the results also suggest the effectiveness of the proposed evaluation protocol in distinguishing and ranking diverse generative models.

For scenarios targeting the task of node classification, we can adopt conditional generation given node labels for better performance, as discussed in Section 2.4. Label-conditional GraphMaker variants, denoted by (cond.) in the table, perform better or no worse than the unconditional ones for about 90% cases. Among the GraphMaker variants, label-conditional GraphMaker-Async achieves the best performance for

Table 2: Evaluation for benchmarking ML models. Best results are in **bold**. OOM stands for out of memory.

| | Cora | | Amazon Photo | | Amazon Computer | |
|---|---|---|---|---|---|---|
| Model | Pearson ↑ | Spearman ↑ | Pearson ↑ | Spearman ↑ | Pearson ↑ | Spearman ↑ |
| ER | $-0.87 \pm 0.03$ | $-0.10 \pm 0.10$ | $0.29 \pm 0.05$ | $0.49 \pm 0.12$ | $0.14 \pm 0.04$ | $0.12 \pm 0.05$ |
| FMF | $0.03 \pm 0.07$ | $-0.25 \pm 0.08$ | $0.13 \pm 0.03$ | $0.37 \pm 0.00$ | $0.43 \pm 0.06$ | $0.66 \pm 0.08$ |
| GAE | $0.04 \pm 0.06$ | $-0.25 \pm 0.08$ | $0.30 \pm 0.07$ | $0.39 \pm 0.03$ | $0.27 \pm 0.05$ | $0.41 \pm 0.03$ |
| VGAE | $-0.02 \pm 0.04$ | $-0.29 \pm 0.07$ | $0.32 \pm 0.14$ | $0.52 \pm 0.03$ | $0.03 \pm 0.11$ | $0.18 \pm 0.15$ |
| GDSS | $-0.44 \pm 0.28$ | $0.03 \pm 0.25$ | OOM | OOM | OOM | OOM |
| EDGE | $0.97 \pm 0.02$ | $0.97 \pm 0.02$ | $0.63 \pm 0.03$ | $0.61 \pm 0.02$ | OOM | OOM |
| GMaker-S | $-0.11 \pm 0.20$ | $0.18 \pm 0.33$ | $0.26 \pm 0.55$ | $0.31 \pm 0.47$ | $-0.09 \pm 0.05$ | $0.12 \pm 0.19$ |
| GMaker-A | $0.97 \pm 0.01$ | $0.93 \pm 0.04$ | **$0.99 \pm 0.00$** | $0.79 \pm 0.07$ | $0.97 \pm 0.01$ | $0.73 \pm 0.05$ |
| GMaker-S (cond.) | $0.96 \pm 0.03$ | **$0.99 \pm 0.02$** | $0.87 \pm 0.05$ | **$0.90 \pm 0.02$** | $0.94 \pm 0.01$ | **$0.90 \pm 0.05$** |
| GMaker-A (cond.) | **$0.98 \pm 0.01$** | $0.95 \pm 0.05$ | $0.90 \pm 0.06$ | $0.87 \pm 0.07$ | **$0.98 \pm 0.02$** | $0.75 \pm 0.12$ |

$18/24 = 75\%$ cases. Meanwhile, it is worth noting that label conditioning narrows the performance gap between GraphMaker-Sync and GraphMaker-Async.

**Utility for Benchmarking ML Models on Node Classification.** Another important scenario of using generated graphs is benchmarking ML models, where industry practitioners aim to select the most effective model architecture from multiple candidates based on their performance on publicly available synthetic graphs and then train a model from scratch on their proprietary real graph. This use case requires the generated graphs to yield reproducible relative performance of the candidate model architectures on the original graph. Following Yoon et al. (2023), for each candidate model architecture, we train and evaluate one model on the original graph for $\text{ACC}(G|G)$ and another on a generated graph for $\text{ACC}(\hat{G}|\hat{G})$. We then report the Pearson/Spearman correlation coefficients between them (Myers et al., 2010). We include all six model architectures for node classification in the set of candidate model architectures.

Table 2 presents the experiment results. Out of 6 cases, unconditional GraphMaker-Async performs better or no worse than all baselines for 5 of them. Overall, the results are consistent with the previous observations in terms of the benefit of asynchronous generation and label conditioning. The only exception is that label-conditional GraphMaker-Sync slightly outperforms label-conditional GraphMaker-Async in terms of Spearman correlation coefficient by a small margin of 0.02 on average. Spearman correlation coefficient examines the reproduction of relative model ranking differences, and is most useful when there exists a substantial performance gap between each pair of candidate models. In practice, when multiple models perform sufficiently similar, the final decision on model selection can involve a tradeoff between cost and performance. In this case, exactly reproducing the relative model ranking differences may be less important than reproducing the relative model performance differences, in which case Pearson correlation coefficient is preferred instead. This applies to our case, as reflected by the raw $\text{ACC}(G|G)$ values reported in Appendix A.3.

### 3.3 Evaluation for structural properties (Q2)

To assess the quality of the generated graph structures, following You et al. (2018b), we report distance metrics for node degree, clustering coefficient, and four-node orbit count distributions. We adopt 1-Wasserstein distance $W_1(x, y)$, where $x, y$ are respectively a graph statistic distribution from the original graph and a generated graph, and a lower value is better. Besides distribution distance metrics, we directly compare a few scalar-valued statistics. Let $M(G)$ be a non-negative-valued statistic, we report $\mathbb{E}_{\hat{G} \sim p_\theta} \left[ M(\hat{G})/M(G) \right]$, where $\hat{G}$ is a generated graph. A value closer to 1 is better. In addition to triangle count, we employ the following metric for measuring the correlations between graph structure and node label (Lim et al., 2021), where a larger value indicates a higher correlation.

$$\hat{h}(\mathbf{A}, \mathbf{Y}) = \frac{1}{C_Y - 1} \sum_{k=1}^{C_Y} \max\left\{ 0, \frac{\sum_{\mathbf{Y}_v=k} |\{u \in \mathcal{N}(v)|\mathbf{Y}_u = \mathbf{Y}_v\}|}{\sum_{\mathbf{Y}_v=k} |\mathcal{N}(v)|} - \frac{|\{v|\mathbf{Y}_v = k\}|}{N} \right\} \quad (4)$$

Table 3: Evaluation for structural properties. Best results are in **bold**. OOM stands for out of memory.

| | Amazon Photo | | | | | |
| | $W_1 \downarrow$ | | | $\mathbb{E}_{\hat{G} \sim p_\theta}\left[M(\hat{G})/M(G)\right] \to 1$ | | |
| Model | Degree ($\times 10^3$) | Cluster ($\times 10$) | Orbit | # Triangle | $\hat{h}(\mathbf{A}, \mathbf{Y})$ | $\hat{h}(\mathbf{A}^2, \mathbf{Y})$ |
|---|---|---|---|---|---|---|
| ER | $0.019 \pm 0.000$ | $0.15 \pm 0.00$ | $1.4 \pm 0.0$ | $0.005 \pm 0.000$ | $0.0013 \pm 0.0002$ | $0.0024 \pm 0.0001$ |
| FMF | $3.8 \pm 0.0$ | $0.15 \pm 0.00$ | $1.4 \pm 0.0$ | $0.0032 \pm 0.0011$ | $0.10 \pm 0.00$ | $0 \pm 0$ |
| GAE | $3.8 \pm 0.0$ | $0.15 \pm 0.00$ | $1.4 \pm 0.0$ | $0.0064 \pm 0.0011$ | $0.062 \pm 0.000$ | $0 \pm 0$ |
| VGAE | $3.8 \pm 0.0$ | $0.15 \pm 0.00$ | $1.4 \pm 0.0$ | $0.0040 \pm 0.0023$ | $0.12 \pm 0.00$ | $0 \pm 0$ |
| GDSS | OOM | OOM | OOM | OOM | OOM | OOM |
| EDGE | $\mathbf{0.0017 \pm 0.0001}$ | $0.1 \pm 0.0$ | $\mathbf{0.060 \pm 0.021}$ | $0.34 \pm 0.04$ | $0.38 \pm 0.02$ | $0.41 \pm 0.03$ |
| GMaker-S | $3.4 \pm 0.0$ | $0.15 \pm 0.00$ | $1.3 \pm 0.0$ | $0.010 \pm 0.001$ | $0.0010 \pm 0.0000$ | $0 \pm 0$ |
| GMaker-A | $0.017 \pm 0.000$ | $\mathbf{0.018 \pm 0.000}$ | $0.41 \pm 0.00$ | $1.5 \pm 0.1$ | $0.88 \pm 0.02$ | $\mathbf{0.93 \pm 0.03}$ |
| GMaker-S (cond.) | $0.074 \pm 0.006$ | $0.13 \pm 0.00$ | $1.0 \pm 0.0$ | $0.1 \pm 0.0$ | $\mathbf{1.1 \pm 0.0}$ | $0.76 \pm 0.02$ |
| GMaker-A (cond.) | $0.0069 \pm 0.0007$ | $0.039 \pm 0.005$ | $0.55 \pm 0.08$ | $\mathbf{0.88 \pm 0.08}$ | $\mathbf{1.1 \pm 0.0}$ | $1.4 \pm 0.0$ |

| | Amazon Computer | | | | | |
| | $W_1 \downarrow$ | | | $\mathbb{E}_{\hat{G} \sim p_\theta}\left[M(\hat{G})/M(G)\right] \to 1$ | | |
| Model | Degree ($\times 10^3$) | Cluster ($\times 10^{-1}$) | Orbit | # Triangle | $\hat{h}(\mathbf{A}, \mathbf{Y})$ | $\hat{h}(\mathbf{A}^2, \mathbf{Y})$ |
|---|---|---|---|---|---|---|
| ER | $\mathbf{0.025 \pm 0.000}$ | $3.0 \pm 0.1$ | $2.2 \pm 0.0$ | $0.022 \pm 0.015$ | $0.00095 \pm 0.00000$ | $0.0029 \pm 0.0000$ |
| FMF | $6.8 \pm 0.0$ | $3.1 \pm 0.1$ | $2.1 \pm 0.0$ | $0.0036 \pm 0.0051$ | $0.067 \pm 0.001$ | $0 \pm 0$ |
| GAE | $7.5 \pm 0.0$ | $3.2 \pm 0.0$ | $2.1 \pm 0.0$ | $0.0036 \pm 0.0051$ | $0.034 \pm 0.000$ | $0 \pm 0$ |
| VGAE | $7.4 \pm 0.0$ | $3.1 \pm 0.0$ | $2.1 \pm 0.0$ | $0.0036 \pm 0.0051$ | $0.083 \pm 0.000$ | $0 \pm 0$ |
| GDSS | OOM | OOM | OOM | OOM | OOM | OOM |
| EDGE | OOM | OOM | OOM | OOM | OOM | OOM |
| GMaker-S | $0.050 \pm 0.000$ | $3.2 \pm 0.0$ | $2.0 \pm 0.0$ | $0 \pm 0$ | $0.0056 \pm 0.0006$ | $0.0099 \pm 0.0001$ |
| GMaker-A | $0.030 \pm 0.000$ | $2.1 \pm 0.0$ | $\mathbf{0.86 \pm 0.00}$ | $1.9 \pm 0.2$ | $\mathbf{0.84 \pm 0.01}$ | $0.79 \pm 0.09$ |
| GMaker-S (cond.) | $0.26 \pm 0.03$ | $2.7 \pm 0.1$ | $1.5 \pm 0.1$ | $0.072 \pm 0.005$ | $1.2 \pm 0.0$ | $0.54 \pm 0.01$ |
| GMaker-A (cond.) | $0.21 \pm 0.00$ | $\mathbf{2.0 \pm 0.3}$ | $1.2 \pm 0.0$ | $\mathbf{0.46 \pm 0.08}$ | $1.3 \pm 0.0$ | $\mathbf{1.2 \pm 0.0}$ |

where $C_Y$ is the number of node classes, and $\mathcal{N}(v)$ consists of the neighboring nodes of $v$. We also report the metric for two-hop correlations, denoted by $\hat{h}(\mathbf{A}^2, \mathbf{Y})$. The computation of clustering coefficients, orbit counts, and triangle count may be costly for the large generated graphs. So, in such cases, we sample an edge-induced subgraph with the same number of edges from all generated graphs.

Table 3 displays the evaluation results on Amazon Photo and Amazon Computer. See Appendix A.5 for the results on Cora. None of the models consistently outperforms the rest. In terms of the number of cases where they achieve the best performance, the best model is label-conditional GraphMaker-Async with 6 out of 18. The degree-guidance mechanism adopted by EDGE is effective in capturing local structural patterns and leads to the best performance for degree $W_1$ and orbit $W_1$ on Cora and Amazon Photo. Across the conditional and unconditional cases, GraphMaker-Async outperforms GraphMaker-Sync in $W_1$ metrics and triangle count for $21/24 = 87.5\%$ cases, demonstrating a superior capability in capturing edge dependencies. In consistent with the evaluation using discriminative models, label conditioning yields a better performance for $26/36 \approx 72\%$ cases.

**Diversity.** Based on evaluation with statistics and discriminative models, we also show that GraphMaker is capable of generating diverse and hence novel graphs in Appendix A.4.

## 3.4 Evaluation for graph recovery (Q3)

To quantify the recovery of the original graph by the synthetic graphs, we evaluate their similarity. Precisely measuring the similarity between two graph structures requires solving the subgraph isomorphism problem, which is known to be NP-complete (Cook, 1971). This challenge is further complicated by the presence of numerous node attributes. Therefore, we propose local node-centered comparisons based on the node

Table 4: Evaluation for graph similarity on Cora. The most similar results are in **bold**.

| Model | Attribute ($\times 10^{-2}$) | 1-hop ($\times 10^{-2}$) | 2-hop ($\times 10^{-3}$) |
|---|---|---|---|
| ER | $1.6 \pm 0.0$ | $1.8 \pm 0.0$ | $9.7 \pm 0.2$ |
| FMF | $1.6 \pm 0.0$ | $\mathbf{1.2 \pm 0.0}$ | $7.7 \pm 0.5$ |
| GAE | $1.6 \pm 0.0$ | $\mathbf{1.2 \pm 0.0}$ | $7.7 \pm 0.4$ |
| VGAE | $1.6 \pm 0.0$ | $\mathbf{1.2 \pm 0.0}$ | $7.8 \pm 0.4$ |
| GDSS | $46.7 \pm 0.2$ | $48.9 \pm 0.3$ | $46.3 \pm 0.1$ |
| EDGE | $\mathbf{0.9 \pm 0.0}$ | $1.3 \pm 0.0$ | $4.0 \pm 0.0$ |
| GMaker-S | $1.5 \pm 0.0$ | $1.8 \pm 0.1$ | $9.2 \pm 0.3$ |
| GMaker-A | $1.3 \pm 0.0$ | $1.5 \pm 0.1$ | $6.4 \pm 0.1$ |
| GMaker-S (cond.) | $1.3 \pm 0.0$ | $1.5 \pm 0.0$ | $6.8 \pm 0.1$ |
| GMaker-A (cond.) | $\mathbf{0.9 \pm 0.0}$ | $\mathbf{1.2 \pm 0.0}$ | $\mathbf{3.7 \pm 0.1}$ |

Table 5: Ablation study for node-personalized label-conditional models on Cora. Better results are in **bold**.

| Model | PE | $W_1 \downarrow$ | | | $\mathbb{E}_{\hat{G} \sim p_\theta}\left[M(\hat{G})/M(G)\right] \to 1$ | | |
|---|---|---|---|---|---|---|---|
| | | Degree | Cluster | Orbit | # Triangle | $\hat{h}(\mathbf{A}, \mathbf{Y})$ | $\hat{h}(\mathbf{A}^2, \mathbf{Y})$ |
| GMaker-S | | 0.62 | $\underline{\mathbf{23}}$ | $\underline{\mathbf{1.3}}$ | $\underline{\mathbf{0.071}}$ | $\underline{\mathbf{0.92}}$ | 0.95 |
| (cond.) | ✓ | $\underline{\mathbf{0.53}}$ | $\underline{\mathbf{23}}$ | 1.4 | 0.06 | 0.91 | $\underline{\mathbf{0.96}}$ |
| GMaker-A | | 1.5 | 9.1 | 0.61 | $\underline{\mathbf{1.4}}$ | $\underline{\mathbf{1.1}}$ | $\underline{\mathbf{1.1}}$ |
| (cond.) | ✓ | $\underline{\mathbf{1.2}}$ | $\underline{\mathbf{7.6}}$ | $\underline{\mathbf{0.49}}$ | 1.6 | $\underline{\mathbf{1.1}}$ | 1.2 |

| Model | PE | Node Classification $\to 1$ | | | | | | Link Prediction $\to 1$ | | |
|---|---|---|---|---|---|---|---|---|---|---|
| | | MLP | 1-SGC | L-SGC | L-GCN | 1-APPNP | L-APPNP | CN | 1-GAE | L-GAE |
| GMaker-S | | $\underline{\mathbf{1.00}}$ | $\underline{\mathbf{0.93}}$ | $\underline{\mathbf{1.01}}$ | $\underline{\mathbf{1.01}}$ | $\underline{\mathbf{0.99}}$ | 1.01 | $\underline{\mathbf{1.00}}$ | $\underline{\mathbf{0.98}}$ | $\underline{\mathbf{0.98}}$ |
| (cond.) | ✓ | 1.02 | 0.89 | $\underline{\mathbf{0.99}}$ | $\underline{\mathbf{0.99}}$ | 0.95 | $\underline{\mathbf{1.00}}$ | $\underline{\mathbf{1.00}}$ | 0.97 | $\underline{\mathbf{0.98}}$ |
| GMaker-A | | / | $\underline{\mathbf{0.93}}$ | $\underline{\mathbf{1.00}}$ | $\underline{\mathbf{1.00}}$ | $\underline{\mathbf{0.96}}$ | 1.01 | $\underline{\mathbf{1.00}}$ | $\underline{\mathbf{0.98}}$ | $\underline{\mathbf{1.00}}$ |
| (cond.) | ✓ | / | $\underline{\mathbf{0.93}}$ | $\underline{\mathbf{1.00}}$ | 0.99 | 0.95 | $\underline{\mathbf{1.00}}$ | $\underline{\mathbf{1.00}}$ | $\underline{\mathbf{0.98}}$ | 0.99 |

labels and attributes. Given a node $v$ in the original graph, we define its most similar counterpart $\pi(v)$ in a synthetic graph as follows:

$$\pi(v) = \underset{\left\{\hat{v} \in \hat{\mathcal{V}} | \hat{\mathbf{Y}}_{\hat{v}} = \mathbf{Y}_v\right\}}{\arg\min} \|\mathbf{X}_v - \hat{\mathbf{X}}_{\hat{v}}\|_1, \tag{5}$$

where $\hat{\mathcal{V}}$, $\hat{\mathbf{X}}$, and $\hat{\mathbf{Y}}$ are the node set, node attributes, and node labels in a synthetic graph, respectively. We then report the overall attribute discrepancy with:

$$\frac{1}{N} \cdot \frac{1}{F} \sum_{v \in \mathcal{V}} \|\mathbf{X}_v - \hat{\mathbf{X}}_{\pi(v)}\|_1. \tag{6}$$

To account for structures, we extend the comparison to $K$-hop neighbors $\mathcal{N}^{(K)}(v)$ and $\hat{\mathcal{N}}^{(K)}(\pi(v))$:

$$\frac{1}{N} \cdot \frac{1}{F} \sum_{v \in \mathcal{V}} \min_{\substack{u \in \mathcal{N}^{(K)}(v), \\ \hat{u} \in \hat{\mathcal{N}}^{(K)}(\pi(v))}} \|\mathbf{X}_u - \hat{\mathbf{X}}_{\hat{u}}\|_1. \tag{7}$$

Table 4 presents the evaluation results on Cora. While label-conditional GraphMaker-Async generally produces the most similar synthetic graphs, its associated metrics remain at a similar order of magnitude compared to other approaches, suggesting that GraphMaker does not merely replicate the original graph.

## 3.5 Ablation study for node personalization (Q4)

We perform an ablation study to see if capturing each node's personalized behavior helps improve the graph generation quality. We adopt a state-of-the-art positional encoding (PE) method RFP (Eliasof et al., 2023). It uses random node features after a certain rounds of message passing operations as PEs, where

the random initial features can be viewed as identity information to distinguish a node from the others. In our experiments, for each noisy graph either during training or generation, we compute PEs based on the current noisy graph structure and then use PEs as extra node attributes. This essentially assists the model in encoding more personalized behaviors of nodes. We examine the effects of such personalization on label-conditional GraphMaker for Cora. Table 5 presents the evaluation results for structural properties and ML utility. For $22/29 \approx 76\%$ cases, GraphMaker variants without PE perform better than or comparable to GraphMaker variants with PE, which suggests that personalized node behaviors may not be really useful to learn a graph generative model, at least in our downstream evaluation.

## 4 Further related works

We have reviewed the diffusion-based graph generative models in Section 1. Here, we review some other non-diffusion deep generative models and some recent improvement efforts on synthetic graph data evaluation. We leave a review on classical graph generative models to Appendix A.6.

**Non-diffusion deep generative models of large graphs.** GAE and VGAE (Kipf & Welling, 2016) extend AE and VAE (Kingma & Welling, 2014; Rezende et al., 2014) respectively for reconstructing and generating the structure of a large attributed graph. NetGAN (Bojchevski et al., 2018) extends WGAN (Arjovsky et al., 2017) for graph structure generation by sequentially generating random walks. GraphRNN (You et al., 2018b) and Li et al. (2018) propose auto-regressive models that generate graph structures by sequentially adding individual nodes and edges. GRAN (Liao et al., 2019) introduces a more efficient auto-regressive model that generates graph structures by progressively adding subgraphs. CGT (Yoon et al., 2023) tackles a simplified version of the large attributed graph generation problem that we consider. It clusters real node attributes during data pre-processing and generates node-centered ego-subgraphs (essentially trees) with a single categorical node label that indicates the attribute cluster.

**Evaluation of generated graphs with discriminative models.** GraphWorld (Palowitch et al., 2022) is a software for benchmarking discriminative models on synthetic attributed graphs, but it does not compare synthetic graphs against a real graph in benchmarking. CGT (Yoon et al., 2023) uses generated subgraphs to benchmark GNNs. It evaluates one model for node classification on the original graph and another on generated subgraphs, and then measures the performance correlation/discrepancy of the models. Our work instead is the first effort to train models on a generated graph and then evaluate them on the original graph. Outside the graph domain, CAS employs discriminative models to evaluate the performance of conditional image generation (Ravuri & Vinyals, 2019).

## 5 Conclusion

We propose GraphMaker, a diffusion model capable of generating large attributed graphs, along with a novel evaluation protocol that assesses generation quality by training models on generated graphs and evaluating them on real ones. Overall, GraphMaker achieves better performance compared to baselines, for both existing metrics and the newly proposed evaluation protocol. Potential future works include extending GraphMaker for larger graphs with 1M+ nodes, continuous-valued node attributes, node labels that indicate node anomalies for anomaly detection, etc.

### Broader Impact Statement

The current manuscript primarily focuses on building a model for generating high-quality synthetic graphs. However, for privacy-preserving synthetic data sharing, there is likely a privacy-utility trade-off that warrants more thorough investigation in future work. Additionally, the misuse of GraphMaker and the generated synthetic graphs may lead to the spread of misinformation. Therefore, the release of a synthetic graph should be accompanied by a statement indicating its source.

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

# A  Appendix

## A.1  Dataset statistics

Table 6 presents the detailed dataset statistics.

Table 6: Dataset statistics. For $|\mathcal{E}|$, we add reverse edges and then remove duplicate edges.

| Dataset | $|\mathcal{V}|$ | $|\mathcal{E}|$ | # labels | # attributes |
|---|---|---|---|---|
| Cora | $2,708$ | $10,556$ | 7 | $1,433$ |
| Amazon Photo | $7,650$ | $238,163$ | 8 | 745 |
| Amazon Computer | $13,752$ | $491,722$ | 10 | 767 |

## A.2  Details for baseline experiments

For a fair comparison, we augment the input of feature-based MF, GAE, and VGAE to include node attributes and label. During generation, we first sample node attributes and node label, and then pass them along with an empty graph with self-loops only for link prediction.

Table 7: Evaluation with MLP. Best results are in **bold**. Highest results are underlined.

| Model | Cora | Amazon Photo | Amazon Computer |
|---|---|---|---|
| $p(\mathbf{Y}) \prod_v \prod_f p(\mathbf{X}_{v,f})$ | $0.57 \pm 0.02$ | $0.20 \pm 0.08$ | $0.16 \pm 0.07$ |
| $p(\mathbf{Y}) \prod_v \prod_f p(\mathbf{X}_{v,f}|\mathbf{Y}_v)$ | $\underline{1.10 \pm 0.04}$ | $\mathbf{0.97 \pm 0.04}$ | $0.97 \pm 0.02$ |
| GraphMaker-Sync (conditional) | $\mathbf{1.00 \pm 0.04}$ | $\mathbf{0.97 \pm 0.01}$ | $0.90 \pm 0.01$ |
| GraphMaker-Async (conditional) | $1.03 \pm 0.04$ | $\underline{1.05 \pm 0.01}$ | $\mathbf{1.00 \pm 0.05}$ |

Table 8: Performance of discriminative models trained and evaluated on the original graph.

| Dataset | Node Classification (ACC) | | | | | | Link Prediction (AUC) | | |
|---|---|---|---|---|---|---|---|---|---|
| | MLP | 1-SGC | L-SGC | L-GCN | 1-APPNP | L-APPNP | CN | 1-GAE | L-GAE |
| Cora | 0.546 | 0.764 | 0.786 | 0.807 | 0.780 | 0.810 | 0.707 | 0.930 | 0.925 |
| Amazon Photo | 0.699 | 0.619 | 0.638 | 0.895 | 0.900 | 0.913 | 0.934 | 0.952 | 0.958 |
| Amazon Computer | 0.615 | 0.554 | 0.542 | 0.780 | 0.793 | 0.774 | 0.925 | 0.942 | 0.911 |

For EDGE, we adopt the hyperparameters it used for generating the structure of Cora. The original EDGE paper generates molecules by first sampling atom types and then generating a graph structure conditioned on the sampled atom types. We extend it for generating a large attributed graph by first generating node attributes and label and then generating the graph structure conditioned on them. To generate attributes and label, we use the attribute generation component in the unconditional GraphMaker-Async.

For GDSS, we adopt the hyperparameters provided in the open source codebase and the S4 integrator proposed in the original paper to simulate the system of reverse-time SDEs. As GDSS directly denoises the dense adjacency matrices, it needs to decide edge existence based on a threshold to obtain the final graph structure. We choose a threshold so that the number of generated edges is the same as the number of edges in the original graph.

### A.3  Additional details for evaluation with discriminative models

To evaluate the quality of the generated attributes, we train one MLP on the real training node attributes and label, and another MLP on the generated training node attributes and label. We then evaluate the two models on the real test node attributes and label to obtain two performance metrics $\mathrm{ACC}(G|G)$ and $\mathrm{ACC}(G|\hat{G})$. If the ratio $\mathrm{ACC}(G|\hat{G})/\mathrm{ACC}(G|G)$ is close to one, then the generated data is considered as having a utility similar to the original data for training the particular model.

For all evaluation conducted in this paper with discriminative models, we design a hyperparameter space specific to each discriminative model, and implement a simple AutoML pipeline to exhaustively search through a hyperparameter space for the best trained model.

Table 7 presents the results for node classification with MLP. The conditional empirical distribution $p(\mathbf{Y}) \prod_v \prod_f p(\mathbf{X}_{v,f}|\mathbf{Y}_v)$ consistently outperforms the unconditional variant $p(\mathbf{Y}) \prod_v \prod_f p(\mathbf{X}_{v,f})$.

Table 8 details the performance of the discriminative models trained and evaluated on the original graph.

### A.4  Evaluation for diversity and novelty

To study the diversity of the graphs generated by GraphMaker, we generate 50 graphs with each conditional GraphMaker variant and make histogram plots of metrics based on them. For structure diversity, we report $W_1$ for node degree distribution. For node attribute and label diversity, we train an MLP on the original graph and report its accuracy on generated graphs. Figure 2 presents the histogram plots for Cora, which demonstrates that GraphMaker is capable of generating diverse and hence novel graphs.

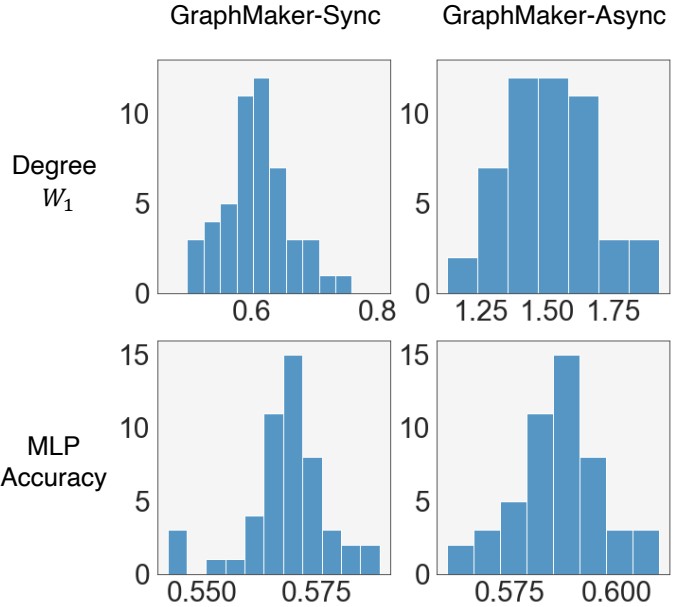

Figure 2: Histogram plots of metrics, which demonstrate the diversity of the generated graphs.

Table 9: Evaluation for structural properties. Best results are in **bold**.

| Model | Cora | | | | | |
|---|---|---|---|---|---|---|
| | $W_1 \downarrow$ | | | $\mathbb{E}_{\hat{G} \sim p_\theta}\left[ M(\hat{G})/M(G) \right] \to 1$ | | |
| | Degree ($\times 10^3$) | Cluster ($\times 10$) | Orbit | # Triangle | $\hat{h}(\mathbf{A}, \mathbf{Y})$ | $\hat{h}(\mathbf{A}^2, \mathbf{Y})$ |
| ER | $0.001 \pm 0.000$ | $2.4 \pm 0.0$ | $1.6 \pm 0.0$ | $0.0066 \pm 0.0002$ | $0.0068 \pm 0.0018$ | $0.092 \pm 0.003$ |
| FMF | $1.3 \pm 0.0$ | $2.4 \pm 0.0$ | $1.6 \pm 0.0$ | $0.0063 \pm 0.0025$ | $0.11 \pm 0.00$ | $0 \pm 0$ |
| GAE | $1.3 \pm 0.0$ | $2.4 \pm 0.0$ | $1.6 \pm 0.0$ | $0.0071 \pm 0.0018$ | $0.11 \pm 0.00$ | $0 \pm 0$ |
| VGAE | $1.4 \pm 0.0$ | $2.4 \pm 0.0$ | $1.6 \pm 0.0$ | $0.0060 \pm 0.0019$ | $0.10 \pm 0.00$ | $0 \pm 0$ |
| GDSS | $0.00089 \pm 0.00001$ | $2.4 \pm 0.0$ | $1.5 \pm 0.0$ | $0.0076 \pm 0.0020$ | $0.013 \pm 0.003$ | $0.084 \pm 0.006$ |
| EDGE | $\underline{\mathbf{0.00058 \pm 0.00002}}$ | $1.7 \pm 0.0$ | $\underline{\mathbf{0.27 \pm 0.05}}$ | $0.32 \pm 0.02$ | $0.82 \pm 0.08$ | $0.77 \pm 0.09$ |
| GMaker-S | $0.0011 \pm 0.0001$ | $2.4 \pm 0.0$ | $1.7 \pm 0.0$ | $0.0037 \pm 0.0019$ | $0.072 \pm 0.008$ | $0.20 \pm 0.01$ |
| GMaker-A | $0.0016 \pm 0.0001$ | $1.5 \pm 0.1$ | $0.56 \pm 0.00$ | $\underline{\mathbf{0.82 \pm 0.06}}$ | $0.86 \pm 0.02$ | $0.75 \pm 0.06$ |
| GMaker-S (cond.) | $0.00061 \pm 0.00006$ | $2.3 \pm 0.0$ | $1.4 \pm 0.1$ | $0.059 \pm 0.014$ | $\underline{\mathbf{0.91 \pm 0.02}}$ | $\underline{\mathbf{0.94 \pm 0.04}}$ |
| GMaker-A (cond.) | $0.0015 \pm 0.0002$ | $\underline{\mathbf{0.90 \pm 0.09}}$ | $0.77 \pm 0.18$ | $1.5 \pm 0.1$ | $1.1 \pm 0.0$ | $1.2 \pm 0.0$ |

## A.5 Evaluation for structural properties on Cora

See Table 9.

## A.6 Review on classic graph generative models

ER (Erdős & Rényi, 1959) generates graph structures with a desired number of nodes and average node degree. SBM (Holland et al., 1983) produces graph structures with a categorical cluster label per node that meet target inter-cluster and intra-cluster edge densities. BA (Barabási & Albert, 2002) generates graph structures whose node degree distribution follows a power law. Chung-Lu (Chung & Lu, 2002) generates graphs with a node degree distribution that equals to a pre-specified node degree distribution in expecta-

Table 10: Hyperparameters for GraphMaker-Sync.

|  | Hyperparameter | Cora | Amazon Photo | Amazon Computer |
|---|---|---|---|---|
| Node attribute prediction | Hidden size for time step representations | 32 | 16 | 16 |
|  | Hidden size for node representations | 512 | 512 | 512 |
|  | Hidden size for node label representations | 64 | 64 | 64 |
|  | Number of MPNN layers | 2 | 2 | 2 |
|  | Learning rate | 0.001 | 0.001 | 0.001 |
|  | Optimizer | AMSGrad | AMSGrad | AMSGrad |
| Link prediction | Hidden size for time step representations | 32 | 16 | 16 |
|  | Hidden size for node representations | 512 | 512 | 512 |
|  | Hidden size for node label representations | 64 | 64 | 64 |
|  | Hidden size for edge representations | 128 | 128 | 128 |
|  | Number of MPNN layers | 2 | 2 | 2 |
|  | Learning rate | 0.0003 | 0.0003 | 0.0003 |
|  | Optimizer | AMSGrad | AMSGrad | AMSGrad |
|  | Batch size | 16384 | 524288 | 2097152 |
| Others | Number of diffusion steps | 3 | 3 | 3 |
|  | Patience for early stop | 20 | 15 | 15 |
|  | Maximal gradient norm for clipping | 10 | 10 | 10 |

tion. Kronecker graph model (Leskovec et al., 2010) generates realistic graphs recursively by iterating the Kronecker product.

### A.7 Implementation details

We implement our work based on PyTorch (Paszke et al., 2019) and DGL (Wang et al., 2019).

### A.8 Hyperparameters

We perform an early stop when the evidence lower bound (ELBO) stops getting improved for a pre-specified number of consecutive evaluations (patience). Table 10 lists the hyperparameters for GraphMaker-Sync. Table 11 lists the hyperparameters for GraphMaker-Async.

Table 11: Hyperparameters for GraphMaker-Async.

| | Hyperparameter | Cora | Amazon Photo | Amazon Computer |
|---|---|---|---|---|
| Node attribute prediction | Hidden size for time step representations | 32 | 16 | 16 |
| | Hidden size for node representations | 512 | 512 | 1024 |
| | Hidden size for node label representations | 64 | 64 | 64 |
| | Number of linear layers | 2 | 2 | 2 |
| | Dropout | 0.1 | 0 | 0 |
| | Number of diffusion steps | 6 | 6 | 7 |
| | Learning rate | 0.001 | 0.001 | 0.001 |
| | Optimizer | AMSGrad | AMSGrad | AMSGrad |
| Link prediction | Hidden size for time step representations | 32 | 16 | 16 |
| | Hidden size for node representations | 512 | 512 | 512 |
| | Hidden size for node label representations | 64 | 64 | 64 |
| | Hidden size for edge representations | 128 | 128 | 128 |
| | Number of MPNN layers | 2 | 2 | 2 |
| | Dropout | 0.1 | 0 | 0 |
| | Number of diffusion steps | 9 | 9 | 9 |
| | Learning rate | 0.0003 | 0.0003 | 0.0003 |
| | Optimizer | AMSGrad | AMSGrad | AMSGrad |
| | Batch size | 16384 | 262144 | 2097152 |
| Others | Patience for early stop | 20 | 15 | 15 |
| | Maximal gradient norm for clipping | 10 | 10 | 10 |

