# OpenReview forum: "GraphMaker: Can Diffusion Models Generate Large Attributed Graphs?"
_TMLR — Accepted by TMLR_

### Review · Reviewer_6cRc · 2024-05-20

**Summary Of Contributions:**

This paper proposes a novel graph generation model GraphMaker. Different from previous graph generation models, GraphMaker mainly focuses on the generation of large attribute graphs. This model is based on the diffusion model and asynchronous generation method to generate large attribute graphs, and introduces a new evaluation method to prove the effectiveness of the generated graphs. This provides ideas for future researchers to explore larger graph generation tasks.

**Audience:**

Yes

**Claims And Evidence:**

Yes

**Requested Changes:**

As pointed out in the weaknesses part, the authors should use more realistic and challenging datasets.

**Strengths And Weaknesses:**

## Strengths

1.	GraphMaker adopts an asynchronous approach to generate node attributes and graph structures, which more effectively captures the complex correlations between attributes and structures. Compared to synchronous generation, asynchronous generation performs better at learning and simulating patterns in real-world large property graphs.
2.	GraphMaker adopts the strategy of edge mini-batch generation and designs a new information transfer neural network (MPNN) to encode data efficiently, thereby improving the model's ability to process large-scale graphs.
3.	This paper proposes a novel evaluation protocol to evaluate the data utility of generated graphs by training machine learning models on the generated graphs and evaluating their performance on the original graphs. This method allows for a more efficient check of the quality of the generated graphs.

## Weaknesses

1.	Table 1 shows that the graph generated by the model can be a good substitute for the original graph, so the two have similar effects in downstream tasks. Table 2 also shows that the generated graph correlates very well with the original graph. However, the performance in Table1 and Table2 can only show that the generated graph is very similar to the original graph, and does not clearly show that the generated graph can bring additional benefits to downstream tasks. For example, when the generated graph is used together with the original graph, the model's performance on downstream tasks cannot be compared to the performance of the original graph alone.
2.	The data sets used in the experiment are relatively small. For example, the Cora dataset is a smaller graph dataset that was proposed in 2008. Experiments require a larger-scale Dataset to prove the effectiveness of the work. For example, the OGB dataset proposed in article " Open Graph Benchmark: Datasets for Machine Learning on Graphs" can be used to conduct larger-scale experiments to prove the model's effectiveness.

---

> ### Author Response · Authors · 2024-08-09
> **Response to Reviewer 6cRc**
>
> Thank you for carefully reviewing our manuscript and providing valuable suggestions.
>
> **Q1**: The empirical studies can only show that the generated graph is a good substitute for the original graph. Can generated graphs bring additional benefits for downstream tasks?
>
> Privacy-preserving synthetic data sharing is a key motivation for GraphMaker, which requires generating a synthetic graph that can serve as a good substitute for the original graph. Depending on the privacy requirement, practitioners can then make privacy-quality tradeoffs by training GraphMaker with privacy-preserving approaches like DPSGD [1]. We agree that exploring the use of synthetic graphs to improve downstream model performance is a valuable direction for future research.
>
> **Q2**: Are the graphs studied sufficiently large?
>
> There are indeed larger graph datasets available for node classification, such as the OGB datasets [2]. Further demonstrating and improving the scalability of GraphMaker would be a meaningful contribution. We will provide an update later if we manage to apply GraphMaker to some OGB datasets before the end of the discussion period. Meanwhile, as highlighted by the other reviewers, the largest graph considered in our manuscript is already about an order of magnitude larger than those used in previous studies on graph generative models.
>
> [1] Abadi et al. Deep learning with differential privacy.
>
> [2] Hu et al. Open Graph Benchmark: Datasets for Machine Learning on Graphs.

---

### Review · Reviewer_pXm7 · 2024-06-13

**Summary Of Contributions:**

The authors propose the GraphMaker diffusion model for generating large attributed graphs. They explore both synchronous (denoising node attributes and edges at all iterations) and asynchronous (first denoise node attributes and then denoise edges) approaches for GraphMaker. The proposed approach scales to 10,000+ nodes, one order of magnitude higher than current diffusion models for graph generation. They perform a comprehensive evaluation that considers multiple different ways of measuring the quality of the generated graph and demonstrate the improvement that GraphMaker provides.

**Audience:**

Yes

**Broader Impact Concerns:**

There is a short broader impact statement that is insufficient. It mentions "protecting privacy", but privacy protection is never evaluated at any point in this paper. This relates to the major issue I mention in Requested Changes regarding the possibility of leaking portions of the original graph data, which should also be mentioned in the statement.

**Claims And Evidence:**

No

**Requested Changes:**

Major issues:
- There is no experiment showing how well the original graph can be recovered from the synthetic graph. I believe that this is an important experiment given the the synthetic graph is generated by starting with the real graph and corrupting it. A graph that is less corrupted should score highly on the current experiment metrics but also have a higher probability of leaking out the original graph itself. This should probably also be discussed in the broader impact statement.

Minor issues:
- Presentation in some of the tables is confusing, such as the scientific notation used in Tables 3 and 4, as well as some tables in the appendix. I have never seen quantities denoted using notations such as $7.0e^{-3}$, only $7.0 \times 10^{-3}$ or the compact version 7.0e-3 without exponent. Ideally, the exponents could be taken out of the table entries itself by using a common exponent, described in the table heading. For example, # Triangle ($\times 10^{-3}$). If the authors insist on using their current notation, then at least explain it in the table caption.
- Last paragraph of introduction: insert missing word "of": 80% of cases and 50% of cases
- Appendix A.1 is so short that I suggest moving it into the main paper in Section 2.2, where it is referenced. It is not overly complicated and would likely not distract the reader from the main points being discussed.

**Strengths And Weaknesses:**

Strengths:
- Proposed GraphMaker can generate edges, node attributes, and node labels more accurately than current approaches.
- Increases scalability of diffusion models for graph generation by one order of magnitude.
- Well thought out evaluation procedure, considering both prediction tasks on the real graph data and benchmarking using only synthetic graph data. Additionally, the authors also consider evaluation of graph structures.
- Presentation and references have a good balance between recent papers on graph neural networks and more classical network science papers.

Weaknesses:
- Claims the possibility of protecting privacy by generating a synthetic graph, but the synthetic graph could be so close to the actual graph that it doesn't protect privacy at all. See the major issue I describe in Requested Changes.
- Some minor presentation issues, which I mention in Requested Changes.

---

> ### Author Response · Authors · 2024-08-11
> **Response to Reviewer pXm7**
>
> Thank you for your insightful feedback and detailed suggestions. We have updated the manuscript to address the presentation-related issues. Below, we respond to the raised questions and concerns.
>
> **Q1**: As a perfect score can be achieved by the original graph on the current experiment metrics considered, a synthetic graph that scores high may be simply recovering the original graph to a larger extent and consequently leaking more information of the original graph. Have you studied to what extent the synthetic graphs recover the original graphs? This should probably also be discussed in the broader impact statement.
>
> **Response**: This is a valid concern, especially in the context of privacy-preserving synthetic data generation. To address this question, we have conducted additional evaluations on the similarity between the original and synthetic graphs. For a detailed discussion, you can refer to the introduction of Section 3 and Section 3.4. We are open to further improving this evaluation if you have additional suggestions.
>
> Additionally, we agree that this concern should be discussed in the broader impact statement , and we have revised the manuscript accordingly.
>
> **Q2**: The scientific notation $e^{exponent}$ used in tables 3, 4, and appendix is confusing.
>
> **Response**: Previously, we followed the practice in [1]. However, we agree that this notation can be confusing. We have updated the manuscript by taking the exponents out of the table entries and instead using a common exponent in table headings when appropriate.
>
> [1] Liao et al. Efficient Graph Generation with Graph Recurrent Attention Networks.

---

> > ### Comment · Reviewer_pXm7 · 2024-09-07
> >
> > I find the evaluation in Section 3.4 to be reasonable and agree with the authors' assessment that GraphMaker is not simply copying the original graph any more than existing methods. I have no further major concerns.

---

> > > ### Author Response · Authors · 2024-09-07
> > >
> > > Thank you for reading our update. We are glad that we have addressed your concerns, and we are grateful to your efforts in helping us improve the manuscript!

---

### Review · Reviewer_J2L9 · 2024-07-31

**Summary Of Contributions:**

The paper presents GraphMaker, a graph diffusion model following the D3PM framework, centered around a MPNN architecture producing node embeddings from which node attributes and edges are predicted, with training made sub-quadratic by predicting only random subsets of edges at a time.

The authors evaluate this framework on Cora, Amazon Computer and Amazon Photo, comparing against GDSS, EDGE, (V)GAE and non-deep learning Erdős-Rényi and feature-based matrix factorization, and in particular explores training a generative model on a single graph by using the synthetic distribution to train node classifiers. They report competitive performance on most datasets and tasks.

**Audience:**

Yes

**Broader Impact Concerns:**

I think the statement is suitable

**Claims And Evidence:**

No

**Requested Changes:**

### Critical:

0. Add _a lot_ more detail about training setup, evaluation setup, hyperparameters etc. to the appendix or main body to make the results reproducible

1. Include at least one evaluation on e.g. Guacamol or another dataset that EDGE, DiGress etc. were evaluated on to contextualize both your edge implementation and how your network performs on these smaller tasks

2. Perform statistical evaluation on the significance of the differences observed by training multiple seeds with different data subsets/graphs and performing a suitable statistical significance test (e.g. https://en.wikipedia.org/wiki/Welch%27s_t-test). If this is not computationally feasible, at least include analysis on how many would be required and report it (for future reproducibility studies) and come up with a proxy experiment to perform qualitatively

### Nice to haves:

1. Acknowledging that the diffusion model is basically DiGress with an MPNN and subsampling would be nice, but not critical

2. The classification metric $ACC(G|\hat{G})$ is highly reminiscent of https://proceedings.neurips.cc/paper/2019/hash/fcf55a303b71b84d326fb1d06e332a26-Abstract.html which should be mentioned in the related work

3. NetGAN https://arxiv.org/abs/1803.00816 and many early graph generative models trained on only a single network, this should be discussed

**Strengths And Weaknesses:**

### Strengths:

+++ scaling to 13k nodes on a non-autoregressive diffusion model is a great achievement

+++ the ablation study of training on a single graph is very interesting to observe

++ strong results in the studied metrics

### Weaknesses:

--- architecture and training not described in enough detail to reproduce (what is the decoding method involving MLP for the edges? what are the subsampling ratios during training? etc.)

-- limited novelty: the method is basically DiGress with an MPNN and randomized subsets for edge predictions during training

-- no clear definition of what "large" means, no evaluation on a "small" attributed graph (e.g. Guacamol) that would allow to compare the work directly to prior work or comparison to follow-ups to DiGress like SparseDiff which can scale to larger graphs

-- maybe I missed it, but I could not find note of number of seeds used to evaluate, there are no error bars or other attempts to evaluate uncertainty in the evaluation

---

> ### Author Response · Authors · 2024-08-13
> **Response to Reviewer J2L9**
>
> Thank you for reviewing our work and providing constructive feedback.
>
> **Q1**: For the purpose of reproducible results, the manuscript should include more details.
>
> **Response**:
>
> - We have improved the description of the decoder in Section 2.5 for better clarity.
> - We have added appendix A.8 to list hyperparameters, including the minibatch size for link prediction during training.
> - We also plan to release our codebase.
>
> **Q2**: The manuscript should report statistical evaluation on the significance of the differences observed by training with different random seeds.
>
> **Response**: We agree that this is important. We have updated the manuscript to include standard deviations over three random seeds for Cora, which yields reasonably stable results for the best methods. While using multiple random seeds introduces some numerical differences, we observe that this has little influence on the overall conclusions. Due to time and compute constraints, we will update the results for the other datasets in the next version.
>
> **Q3**: Include at least one evaluation on a small attributed graph dataset like GuacaMol that would allow direct comparisons to prior work like DiGress and EDGE.
>
> **Response**: Our approach is specifically designed for large attributed networks, such as social networks and citation graphs, which exhibit patterns significantly different from those in molecule datasets like GuacaMol. For instance, the GDSS paper observes that synchronous generation of node attributes (atom types) and edges (chemical bonds) leads to better performance in molecule generation, which contrasts with our findings for large attributed networks. We do not expect our model to perform well for molecule generation and recommend using models like DiGress for such tasks. Additionally, we included EDGE as a baseline for comparison. However, DiGress is not directly applicable to the scenario we considered due to its high memory consumption.
>
> **Q4**: Ideally the paper should also discuss missing related works like [2] and [3].
>
> **Response**: We have already discussed [2] in Section 4 along with other related works, and we have updated the manuscript to briefly discuss [3].
>
> [1] Jo et al. Score-based Generative Modeling of Graphs via the System of Stochastic Differential Equations.
>
> [2] Bojchevski et al. NetGAN: Generating Graphs via Random Walks.
>
> [3] Ravuri and Vinyals. Classification Accuracy Score for Conditional Generative Models.

---

### Author Response · Authors · 2024-08-08
**General Response**

We are grateful to the reviewers for their insightful feedback and recognition of our contributions, which will help us improve the manuscript.

- Reviewers **J2L9** and **pXm7** recognized the achievement of scaling a non-autoregressive graph diffusion model to 13K nodes, improving upon previous approaches by an order of magnitude.
- Reviewers **pXm7** and **6cRc** appreciated the novel evaluation protocol we proposed, which allows for establishing comprehensive benchmarks for large attributed graph generation in future research.
- Reviewers **J2L9** and **6cRc** found the ablation study results meaningful and supportive of the model design, such as asynchronous generation.
- Reviewers **J2L9** and **pXm7** acknowledged the competitive performance achieved by GraphMaker in empirical studies.

We will further respond to each reviewer separately for the specific questions and revise our manuscript as needed.

---

### Decision · Action_Editor_NXp1 · 2024-09-12

**Recommendation:** Accept with minor revision

**Comment:**

The reviewers' feedback is leaning towards accepting this submission. However, the reviewers requested that the proposed changes (such as the added experiments discussed during the rebuttal phase)  will be added to the paper for it to be accepted.

Therefore, I ask the authors to add these to the final paper as a minor revision.

**Audience:**

All reviewers agree that there is an audience and interest from the community in this submission.

**Claims And Evidence:**

All reviewers agree that the claims and evidence were provided in the submission and later refined during the author-reviewers discussion phase.